# Regulated membrane remodeling by Mic60 controls formation of mitochondrial crista junctions

Manuel Hessenberger[1,2], Ralf M. Zerbes[3,4], Heike Rampelt[3], Séverine Kunz[1,5], Audrey H. Xavier[1,2], Bettina Purfürst[5], Hauke Lilie[6], Nikolaus Pfanner[3,7], Martin van der Laan[8] & Oliver Daumke[1,2]

The mitochondrial contact site and cristae organizing system (MICOS) is crucial for the formation of crista junctions and mitochondrial inner membrane architecture. MICOS contains two core components. Mic10 shows membrane-bending activity, whereas Mic60 (mitofilin) forms contact sites between inner and outer membranes. Here we report that Mic60 deforms liposomes into thin membrane tubules and thus displays membrane-shaping activity. We identify a membrane-binding site in the soluble intermembrane space-exposed part of Mic60. This membrane-binding site is formed by a predicted amphipathic helix between the conserved coiled-coil and mitofilin domains. The mitofilin domain negatively regulates the membrane-shaping activity of Mic60. Binding of Mic19 to the mitofilin domain modulates this activity. Membrane binding and shaping by the conserved Mic60–Mic19 complex is crucial for crista junction formation, mitochondrial membrane architecture and efficient respiratory activity. Mic60 thus plays a dual role by shaping inner membrane crista junctions and forming contact sites with the outer membrane.

[1] Department of Crystallography, Max-Delbrück-Center for Molecular Medicine, Robert-Rössle-Strasse 10, 13125 Berlin, Germany. [2] Institute of Biochemistry, Freie Universität Berlin, Takustrasse 6, 14195 Berlin, Germany. [3] Institute of Biochemistry and Molecular Biology, ZBMZ, Faculty of Medicine, University of Freiburg, 79104 Freiburg, Germany. [4] Faculty of Biology, University of Freiburg, 79104 Freiburg, Germany. [5] Core Facility Electron Microscopy, Max-Delbrück-Center for Molecular Medicine, Robert-Rössle-Strasse 10, 13125 Berlin, Germany. [6] Institute of Biochemistry and Biotechnology, Section of Protein Biochemistry, Martin Luther University Halle-Wittenberg, 06120 Halle/Saale, Germany. [7] BIOSS Centre for Biological Signalling Studies, University of Freiburg, 79104 Freiburg, Germany. [8] Medical Biochemistry and Molecular Biology, Center for Molecular Signaling, PZMS, Saarland University, 66421 Homburg, Germany. Correspondence and requests for materials should be addressed to M.v.d.L. (email: Martin.van-der-laan@uks.eu) or to O.D. (email: oliver.daumke@mdc-berlin.de).

Mitochondria are highly dynamic organelles of endosymbiotic origin that fulfil crucial functions within eukaryotic organisms, including the production of metabolites, the regulation of apoptosis and cellular energy supply[1–5]. Their morphology is characterized by the presence of discrete outer and inner membrane systems[6–9]. Whereas the outer mitochondrial membrane (OM) separates the mitochondria from the cytosol, the inner membrane (IM) comprises the inner boundary membrane (IBM), which is closely apposed to the OM, and cristae membranes. Cristae are tubular or lamellar membranous invaginations of the IM, which can largely differ in size and shape, depending on the metabolic state and cell type[1,7,10]. They exhibit a specific protein composition with a characteristic accumulation of respiratory chain (super-) complexes and $F_1F_0$-ATP synthase oligomers[11–16]. Cristae are linked to the IBM via tubular openings of defined diameter termed crista junctions (CJs)[7,10]. It has been proposed that cristae formation and dynamics may be connected to the mitochondrial fusion and fission machineries[17–19]. Perturbations of these processes are observed in numerous pathologies including neurodegenerative diseases and cancer[1,2,4]. Of note, remodelling of CJs to facilitate cytochrome c release from mitochondria is a key event in the induction of programmed cell death[20].

The identity and properties of the protein machineries controlling CJ formation are only slowly emerging. The mitochondrial contact site and cristae organizing system (MICOS) is an evolutionarily conserved multi-subunit protein assembly of the IM, which localizes to CJs and is crucial for their formation and maintenance. The MICOS complex is composed of at least six subunits in yeast that are termed Mic10, Mic12, Mic19, Mic26, Mic27 and Mic60 (refs 21–25). The mammalian MICOS complex contains at least one additional subunit (Mic25) and a number of MICOS-interacting proteins have been reported (summarized in refs 26,27). Recent studies indicate that CJ formation and maintenance in mammals is controlled by an intricate interplay between MICOS and the dynamin-like GTPase OPA1 (refs 17,18).

Ablation of the MICOS core subunits Mic10 and Mic60 induces profound alterations of mitochondrial architecture: The loss of normal CJ structures leads to a detachment of cristae from the IBM and the accumulation of extended lamellar membrane stacks in the mitochondrial matrix[21–23,28–32]. Overexpression of Mic10 or Mic60 causes strong deformations of cristae membranes and/or CJs[29,33].

Recent studies have demonstrated that MICOS is composed of two distinct subcomplexes centred on the two core components Mic10 and Mic60 (previously termed mitofilin). The Mic10 subcomplex includes the IM-integrated proteins Mic12/QIL1, Mic26 and Mic27, whereas the second subcomplex is formed by Mic60 and the peripheral IM protein Mic19 (together with the Mic19 paralogue Mic25 in mammals)[33–37]. Mic10 is a small integral IM protein with two transmembrane (TM) domains that has recently been shown to oligomerize via conserved glycine motifs, leading to the deformation of membranes in vitro and in vivo[33,38]. Accordingly, a function of Mic10 in bending the mitochondrial IM has been proposed. Mic60 is inserted into the IM via an N-terminal TM domain, yet the major part of the protein is soluble and exposed to the intermembrane space (IMS). A central coiled-coil domain was suggested to act as a protein–protein interaction platform[21,28,29,39]. The conserved C-terminal mitofilin domain of Mic60 is crucial for the integrity of the MICOS complex, but its molecular function has remained unclear[40,41]. Mic60 was shown to interact with partner protein complexes in the OM, like the general protein translocase (TOM complex) and the sorting and assembly machinery (SAM/TOB complex). Thus, the Mic60–Mic19(-Mic25) subcomplex mediates the formation of direct physical contacts between both mitochondrial membrane systems and is thought to anchor CJs to the OM[22,23,26,35,39–43]. Mic60-dependent IM–OM contact sites have recently been implicated in mitochondrial lipid trafficking, a process critical for membrane remodelling[44,45]. The Mic60 partner protein Mic19 comprises a central coiled-coil domain and a C-terminal coiled-coil-helix coiled-coil-helix (CHCH) domain. The latter contains two highly conserved cysteine residues that flank a loop region between two predicted helices and can form a disulfide bond in vivo[46]. The molecular mechanism through which the Mic60–Mic19 subcomplex contributes to the formation of CJs is unknown.

Here we have biochemically characterized the Mic60–Mic19 complex and demonstrate that it can deform liposomes into thin tubules in vitro. We identify a lipid-binding site in the soluble IMS-exposed part of Mic60. The membrane remodelling activity of Mic60 is partially inhibited by its mitofilin domain. Binding of Mic19 to the mitofilin domain releases this inhibition and thus promotes Mic60-mediated membrane remodelling. An intact membrane-binding site in Mic60 is crucial for MICOS complex formation, mitochondrial IM ultrastructure and mitochondrial function. Our findings reveal that Mic60–Mic19 not only anchors CJ structures to the OM, but is also crucial for cristae formation via remodelling of mitochondrial membranes.

## Results

**Mic60 binds and remodels liposomes.** To characterize the molecular function of the MICOS core component Mic60, we expressed a Mic60 variant of the thermophilic fungus Chaetomium thermophilum lacking the N-terminal mitochondrial targeting sequence and TM anchor (Mic60sol, residues 208–691) and purified it to homogeneity (Fig. 1a,b). The equivalent construct of yeast Mic60 tended to aggregate and could therefore not be tested in functional studies. Circular dichroism (CD) measurements of Mic60sol (and all subsequently purified Mic60 and Mic19 variants, see below) indicated a mostly α-helical structure (Supplementary Fig. 1a). In agreement with earlier reports[28,29], Mic60sol was predominantly dimeric in blue native (BN)-PAGE, while a smaller fraction assembled into higher-order oligomers (Fig. 1b and Supplementary Fig. 1b). Sedimentation velocity and equilibrium analytical ultracentrifugation (AUC) experiments confirmed that the predominant species of Mic60sol is a stable dimer (measured relative molecular weight (Mr) $109 \pm 10$ kDa, theoretical Mr of the dimer: 109 kDa) in the concentration range of 0.02–2 mg ml$^{-1}$ (Fig. 1c,d). The measured sedimentation velocity of 2.4 S is very small for a 109 kDa dimeric Mic60sol and indicative of a highly elongated protein structure, consistent with the predicted high coiled-coil content of Mic60sol. At a concentration of 1 mg ml$^{-1}$ and higher, an additional species with an apparent sedimentation velocity of ~4 S appeared (Fig. 1c), probably reflecting the capability of Mic60sol to form higher oligomeric structures.

To test for a possible function of Mic60 in membrane interaction and remodelling, we performed liposome co-sedimentation assays. In these assays, Mic60sol efficiently bound to negatively charged Folch liposomes derived from bovine brain lipids, despite the absence of its N-terminal TM anchor (Fig. 1e, quantified in Supplementary Fig. 1c). The addition of 15, 20 or 30% cardiolipin to the Folch liposomes did not affect membrane association of Mic60sol (Fig. 1e). Strikingly, Mic60sol binding resulted in massive liposome deformation, as indicated by the appearance of long (up to 30 μm), mostly unbranched membrane tubules with a diameter of $90 \pm 30$ nm in negative-stain electron microscopy (EM) (Fig. 1f,g and Supplementary Fig. 2a,b, quantified in Supplementary Fig. 2i–k).

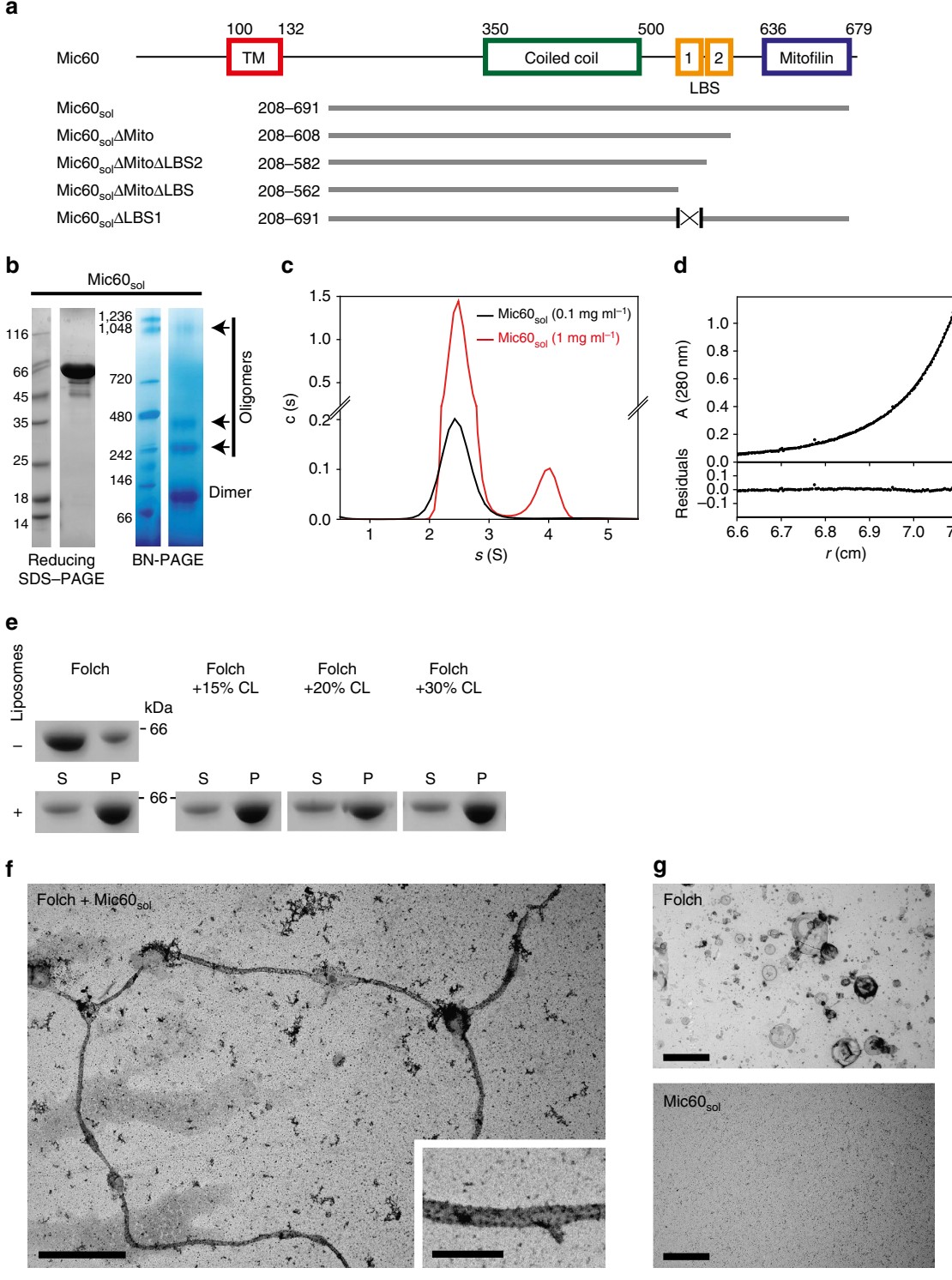

**Figure 1 | Mic60 forms stable dimers and tubulates liposomes.** (**a**, top) Domain architecture of Mic60. TM, transmembrane domain; LBS, lipid-binding site. Constructs used in this study are indicated below. (**b**) Left: SDS–PAGE of Mic60$_{sol}$ under reducing conditions. Right: BN-PAGE analysis of Mic60$_{sol}$ indicates a mainly dimeric species. Arrows indicate higher-order oligomers. (**c**) Sedimentation velocity analysis of Mic60$_{sol}$ at a protein concentration of 0.1 mg ml$^{-1}$ using AUC results in a homogenous species with a sedimentation coefficient $s$(app) = 2.43 ± 0.08S (black). At a concentration of 1 mg ml$^{-1}$, an additional higher oligomeric species (8%) with $s$(app) of ∼4 S appears. (**d**) Sedimentation equilibrium of Mic60$_{sol}$ at a protein concentration of 0.1 mg ml$^{-1}$ yields a molecular mass of Mr = 109 ± 10 kDa, which corresponds to a dimer. The upper panel shows the experimental data (dots) and the fit (solid line), the lower panel displays the deviation of data and fit. (**e**) Folch liposome co-sedimentation assays with Mic60$_{sol}$; S, supernatant; P, pellet: CL, cardiolipin. (**f**) Electron micrograph of negatively stained deformed liposomes after incubation with Mic60$_{sol}$. Electron micrographs of liposomes and Mic60$_{sol}$ protein are shown in (**g**) as controls. Scale bars, 2 μm and 500 nm within the close-up, respectively.

To identify a putative membrane-binding region in the hydrophilic IMS part of Mic60, several C-terminal truncations of Mic60sol were prepared (Fig. 1a). Deletion of the mitofilin domain (Mic60solΔMito, residues 208–608) did not affect dimerization (Supplementary Fig. 1b), membrane binding and tubulation (Fig. 2a). In fact, membrane binding and tubulation appeared slightly enhanced using this construct, with only very few non-deformed liposomes found (Fig. 2a and Supplementary Figs 1c,2c,i,j). Moreover, an increased

appearance of branched tubules was observed (Supplementary Fig. 2k), suggesting that the mitofilin domain regulates Mic60-mediated membrane remodelling. In contrast, a Mic60 variant truncated after the predicted coiled-coil domain (Mic60solΔMitoΔLBS, residues 208–562) did not co-sediment with liposomes (Fig. 2b and Supplementary Fig. 1c), indicating that a lipid-binding site (LBS) may be located in the linker region between the coiled-coil and mitofilin domains (Fig. 1a).

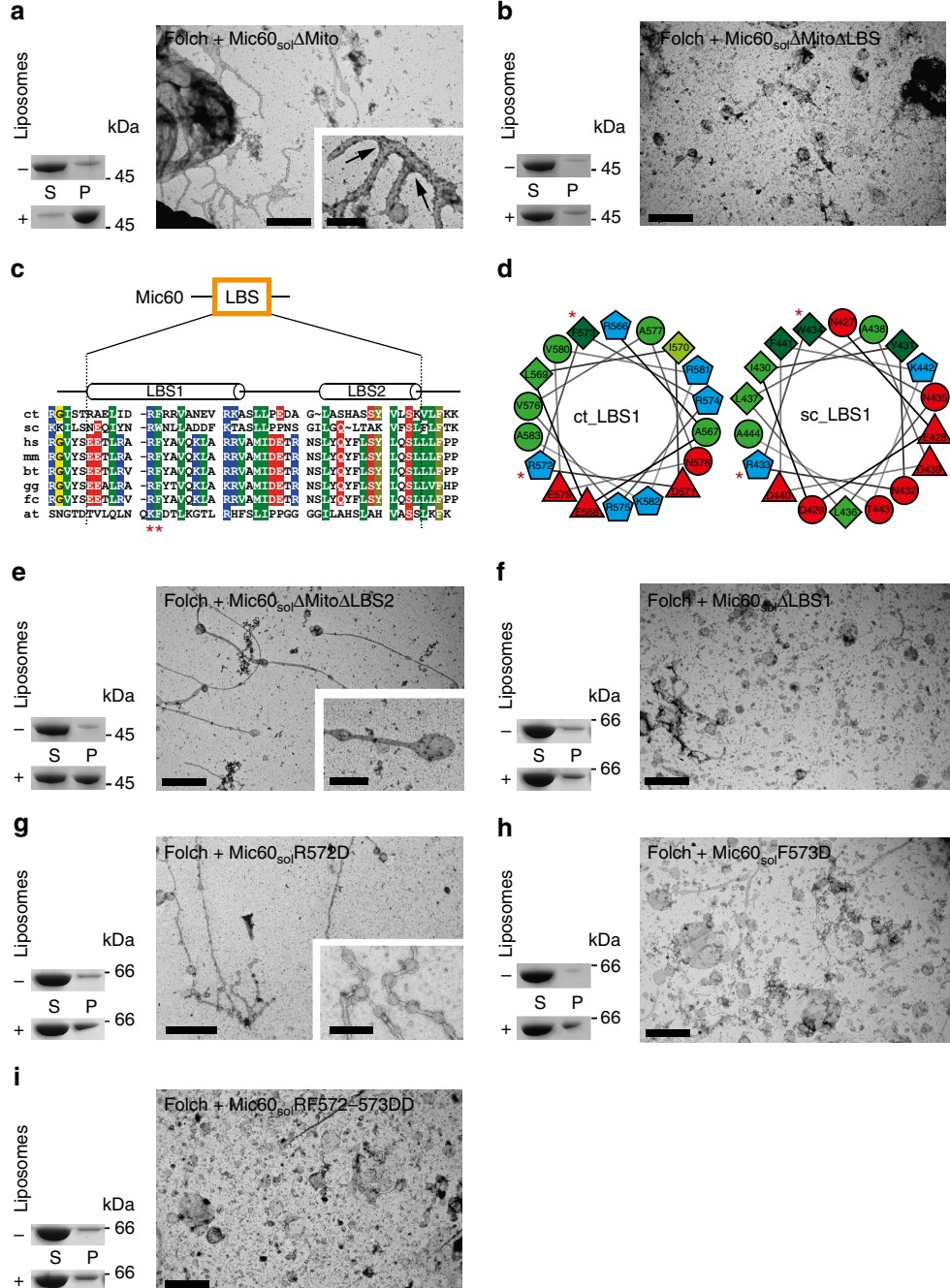

**Figure 2 | Identification of the LBS in Mic60.** (**a,b,e–i**) Liposome sedimentation assays (left) and electron micrographs (right) of Folch liposomes incubated with the indicated Mic60 constructs. Arrows indicate tubule branches. Scale bars, 2 μm and 500 nm within the close-ups, respectively. (**c**) Sequence alignment of the predicted LBS in Mic60, with secondary structure predictions on top. Sequences of Mic60 from *C. thermophilum* (ct), *S. cerevisiae* (sc), *H. sapiens* (hs), *M. musculus* (mm), *B. Taurus* (bt), *G. gallus* (gg), *F. catus* (fc) and *A. thaliana* (at) were aligned. Positively charged residues in blue; negatively charged residues in red, hydrophobic residues in green and uncharged residues in yellow. (**d**) Helical wheel projections of the LBS1 from *C. thermophilum* (ct_LBS1) and *S. cerevisiae* (sc_LBS1) indicate the amphipathic nature of the predicted α-helix. The red asterisks mark residues mutated in this study.

Bioinformatic analysis predicted the presence of two α-helices in this linker region, to which we refer in the following as LBS1 and LBS2 (Figs 1a and 2c). A helical wheel projection revealed a conserved amphipathic character of LBS1 (Fig. 2d), a known feature for membrane inserting helices[47]. The deletion of LBS2 together with the mitofilin domain (Mic60$_{sol}$ΔMitoΔLBS2, residues 208–582) did not affect dimerization (Supplementary Fig. 1b), but reduced membrane interaction of Mic60 (Supplementary Fig. 1c). However, liposome tubulation was still observed (Fig. 2e and Supplementary Fig. 2d). To demonstrate a possible requirement of LBS1 in membrane binding, we specifically removed this region from the soluble Mic60 variant via an internal deletion (Mic60$_{sol}$ΔLBS1) (Fig. 1a). Indeed, this variant was still dimeric (Supplementary Fig. 1b), but neither bound nor tubulated liposomes (Fig. 2f and Supplementary Fig. 1c).

To exactly pinpoint membrane-binding residues, we focused on the conserved residues Arg572 and Phe573. Arg572 represents a potential binding partner for the negatively charged phospholipid head groups, whereas Phe573 may insert into the hydrophobic core of the lipid bilayer. In line with this hypothesis, individual replacements of Arg572 or Phe573 by aspartate (Mic60$_{sol}$R572D or Mic60$_{sol}$F573D) considerably reduced liposome binding (Fig. 2g,h). Whereas the F573D mutant was mostly deficient in liposome tubulation, the R572D variant often induced deformed liposomes in a beads-on-a-string pattern in EM analyses (Fig. 2g,h and Supplementary Fig. 2e,i). If tubules were observed for the latter mutant, they had a larger average diameter of 140 ± 20 nm (Supplementary Fig. 2j), pointing to a reduced potential to deform membranes[48]. Simultaneous replacement of Arg572 and Phe573 with glutamate residues (RF572-573DD variant) completely abolished liposome binding and tubulation, suggesting that these two residues are directly involved in membrane interaction and remodelling (Fig. 2i and Supplementary Fig. 2i).

**Mic19 binds to Mic60 and enhances membrane remodelling.** Mic60 and Mic19 have been shown to interact with each other, forming a MICOS subcomplex[33–37], but little is known about the molecular mechanism of their interaction. We characterized the interaction between the *C. thermophilum* Mic19 and Mic60 proteins using a series of untagged truncation constructs and isothermal titration calorimetry (ITC) measurements (the hexahistidine tag used for purification was proteolytically removed before the binding studies as described in the Methods section) (Fig. 3). In these experiments, full-length Mic19 bound Mic60$_{sol}$ with high affinity ($K_D$ of 170 nM, Fig. 3a,b). Interestingly, we observed a binding number of 0.7, that is, only 70% of the Mic19 molecules bound to Mic60$_{sol}$. Gel filtration analysis of purified Mic19 revealed a mixture of ~70% monomers and 30% dimers that were sensitive to the presence of reducing agents in SDS–PAGE (Supplementary Fig. 3a,b). Accordingly, stable dimer formation was dependent on the presence of two cysteine residues in the CHCH domain, as replacement of both cysteines in the C132S/C143S Mic19 mutant resulted in a purely monomeric variant (Supplementary Fig. 3a,b). This variant showed a reduced binding affinity to Mic60$_{sol}$ ($K_D = 3.5$ μM), but displayed a binding number of 1 (Supplementary Fig. 3c). Furthermore, the double cysteine mutant showed reduced α-helical content in CD measurements compared to its equivalent counterpart (Supplementary Fig. 1a). Thus, our results suggest that formation of an intramolecular disulfide bond in the CHCH domain of Mic19 is critical for the correct folding of the CHCH domain and high-affinity binding to Mic60. In contrast, formation of an intermolecular disulfide bond results in a dimeric species that cannot bind to Mic60.

Deletion of the mitofilin domain in Mic60$_{sol}$ led to a complete loss of Mic19 binding (Fig. 3c), suggesting a critical role of the mitofilin domain for the interaction with Mic19. The isolated mitofilin domain of Mic60 aggregated at concentrations >1 mg ml$^{-1}$, precluding its use in ITC studies. In contrast, the isolated coiled-coil domain of Mic19 was soluble, but did not bind to Mic60$_{sol}$ (Fig. 3d). However, the isolated CHCH domain of Mic19 bound to Mic60$_{sol}$ with a $K_D$ of 390 nM, that is, with a comparably high affinity as full-length Mic19 (Fig. 3e). Thus, our binding studies using untagged proteins indicate that the Mic19–Mic60 subcomplex of MICOS forms via interaction of the CHCH domain of Mic19 and the mitofilin domain of Mic60.

We next asked if Mic19 has a function in modifying Mic60-mediated membrane tubulation. Initially, we analysed membrane binding of Mic19 alone, but found no interaction with Folch liposomes in co-sedimentation assays nor membrane tubulation (Fig. 3f, left; 3g and Supplementary Fig. 1c). However, when Mic60$_{sol}$ was co-incubated with Mic19, co-sedimentation of the Mic60–Mic19 complex with liposomes was observed, suggesting that the two proteins interact when bound to membranes (Fig. 3f, right). Remarkably, the Mic60–Mic19 complex induced liposome tubulation and branching, as Mic60$_{sol}$ΔMito (Fig. 3h and Supplementary Fig. 2f). A similar phenotype was also observed for the Mic19 C132S/C143S mutant indicating that under the conditions of this assay, the Mic19 mutant can still interact with Mic60 at the membrane (Fig. 3i and Supplementary Fig. 2h–k). Addition of the isolated CHCH domain to Mic60$_{sol}$ enhanced membrane tubulation compared to Mic60$_{sol}$ alone, but we observed less branches indicating that other parts of Mic19 also contribute to the functional cooperation with Mic60 (Fig. 3i,j and Supplementary Fig. 2g,i–k).

When closely inspecting EM micrographs of liposomes incubated with the Mic60–Mic19 complex, we often observed small particles reminiscent of membrane remnants (Supplementary Fig. 2f, right). We reasoned that such particles may be generated by fragmentation of membrane tubules caused by enhanced membrane remodelling. Because a quantitative analysis of such fragmented liposomes in negative-stain EM was difficult to interpret, we looked for an alternative approach. Therefore, we applied an established membrane leakage assay[44], in which the release of the fluorescent dye 8-aminonapthalene-1,3,6 trisulfonic acid (ANTS) and its quencher *p*-xylene-bis-pyridinium bromide (DPX) from the interior of liposomes into solution is observed as a time-dependent fluorescence increase[49].

As expected, Mic60$_{sol}$ but not Mic19 alone induced liposome leakage (Fig. 3k) indicative of a membrane remodelling activity. Mic19 greatly enhanced membrane leakage when co-incubated with Mic60$_{sol}$. In contrast, addition of the isolated CHCH domain reduced the membrane leakage activity of Mic60$_{sol}$. Our results are consistent with a model in which Mic60 membrane remodelling activity is enhanced by Mic19. The mitofilin–CHCH domain interaction promotes tubulation but is not sufficient to enhance membrane leakage. We conclude that effective functional cooperation with Mic60 requires Mic19's coiled-coil domain in addition.

**LBS1 plays a critical role in IM ultrastructure and function.** To examine the physiological role of Mic60-mediated membrane binding for mitochondrial membrane architecture, we used the yeast *Saccharomyces cerevisiae* as model organism. We engineered yeast strains carrying chromosomally integrated *MIC60* mutants encoding protein variants with either a deletion of the homologous LBS1 region or an RW433-434DD amino-acid replacement in LBS1, which is equivalent to the RF572-573DD substitution in *C. thermophilum* Mic60 (see alignments in

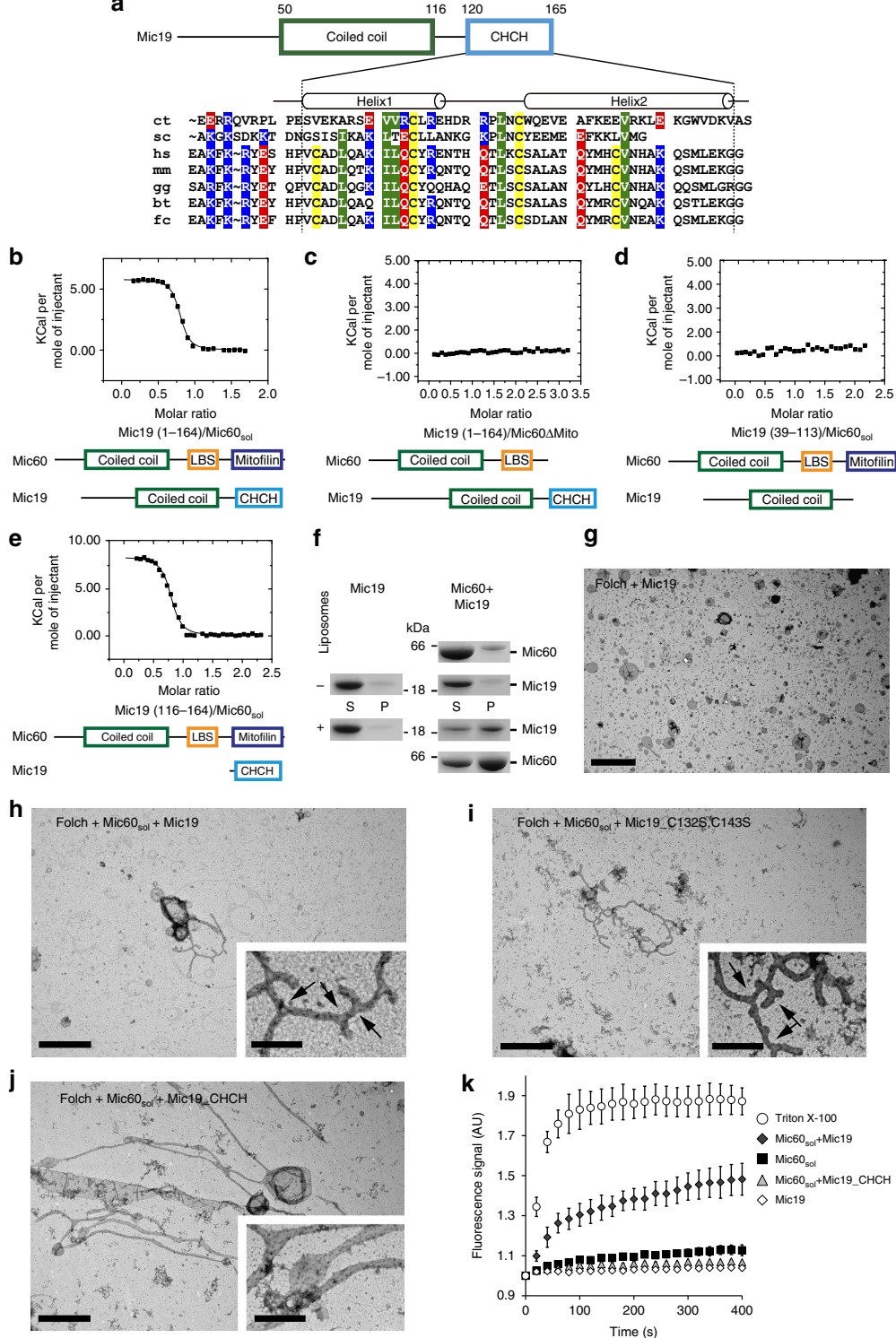

**Figure 3 | Interaction of the CHCH domain of Mic19 and the mitofilin domain of Mic60 controls membrane remodelling.** (**a**) Domain architecture of Mic19; CHCH, coiled-coil-helix coiled-coil-helix domain. (bottom) Sequences of Mic19 from *C. thermophilum* (ct), *S. cerevisiae* (sc), *H. sapiens* (hs), *M. musculus* (mm), *B. Taurus* (bt), *G. gallus* (gg), and *F. catus* (fc) were aligned. Cysteines within the CHCH domain are marked in yellow, positively charged residues in blue; negatively charged residues in red, hydrophobic residues in green. (**b–e**) In ITC experiments, a 450 μM solution of the indicated Mic19 constructs was titrated into 40 μM of the indicated Mic60 constructs at 10 °C, and the resulting heat changes were monitored. Fitted values for **b**: $K_D = 170$ nM ± 20 nM, binding number $n = 0.73 \pm 0.01$. **c**: No binding. **d**: No binding. **e**: $K_D = 390$ nM ± 60 nM, $n = 0.77 \pm 0.01$. (**f**) Liposome co-sedimentation assays of Mic19 alone (left) and Mic19 co-incubated with Mic60$_{sol}$ (right). (**g–h**) Negatively stained electron micrographs of liposomes incubated with (**g**) Mic19 (**h**) Mic60$_{sol}$–Mic19 complex (**i**) Mic60$_{sol}$–Mic19_CHCH and (**j**) Mic60$_{sol}$–Mic19_C132S,C143S. Arrows in the magnification indicate tubule branches. Scale bars, 2 μm and 500 nm within the close-ups, respectively. (**k**) Membrane leakage of Folch liposomes pre-loaded with the fluorescent dye ANTS and its quencher DPX was induced by the addition of the indicated constructs (or 1% Triton X-100 as a control) and monitored by following the time-dependent increase of the fluorescence signal ($n \geq 7$, error bars denote the s.d. of each data point).

Fig. 2c). To enable the purification of the MICOS complex from mutant mitochondria and the co-isolation of MICOS-interacting proteins, a Protein A (ProtA) moiety was fused to the C terminus of the Mic60 variants. It has been reported and was confirmed in our study that the lack of Mic60 leads to strongly reduced Mic19 levels (Supplementary Fig. 4a)[21–23,40]. Mic19 levels were also decreased in mitochondria harbouring Mic60 variants with a defective LBS (Supplementary Fig. 4a) indicating that these variants may not be functional. In contrast, the protein levels of Mic10, Mic26, Mic12 and control proteins, like Tom40, the $F_1F_0$-ATP synthase subunit Atp21 and the respiratory chain complex III subunit Cor1 were not affected.

To test if inactivation of the Mic60 LBS disturbs MICOS integrity, we purified the complex from isolated mitochondria by affinity chromatography with ProtA-tagged wild-type Mic60 and the variants ΔLBS1 and RW433-434DD as bait proteins (Fig. 4a). To make sure that the observed effects were not due to a (partial) loss of Mic19 in mitochondria containing LBS1-defective Mic60 variants, we included Mic60$_{ProtA}$mic19Δ mitochondria for comparison[21,37]. Co-isolation efficiency of other MICOS components together with lipid-binding-deficient Mic60 variants was strongly reduced compared to the co-isolation efficiency with wild-type Mic60$_{ProtA}$, whereas the loss of Mic19 had only moderate effects on the overall integrity of MICOS (Fig. 4a, compare lanes 7–10). Of note, the co-isolation of the OM-binding partner Tom40, the central subunit of the TOM complex, was not reduced with the mutant mitochondria (Fig. 4a), indicating that LBS1-defective Mic60 variants are still able to form IM–OM contact sites. In agreement with the observed MICOS deficiency, yeast strains expressing the lipid-binding-deficient variants of Mic60 showed impaired growth similar to mic60Δ cells on media that require maximal mitochondrial activity (Fig. 4b).

We then expressed Mic19 in the analysed mutant yeast strains from a plasmid leading to the restoration of Mic19 protein levels in mitochondria (Supplementary Fig. 5a). Re-expression of Mic19 in the Mic60_ΔLBS1- and Mic60_RW433-434DD-expressing strains or in a strain lacking the mitofilin domain of Mic60 (ref. 39) neither rescued the assembly of the MICOS complex (Supplementary Fig. 5b) nor the growth phenotype of the mic60 mutant yeast strains on a non-fermentable carbon source (Supplementary Fig. 5c). From these data, we conclude that efficient binding of Mic60 to membranes is crucial for MICOS integrity.

To directly test MICOS functionality in the mic60 mutant strains, we then examined mitochondrial ultrastructure by EM. In Mic60$_{ProtA}$-expressing cells, mitochondria showed a typical membrane architecture with clearly defined cristae that had extensive contacts to the boundary IM via CJs (Fig. 4c, quantified as CJs/mitochondrial section in Fig. 4g). Deletion of the entire MIC60 gene or removal of only the Mic60 mitofilin domain[40] led to a grossly aberrant mitochondrial ultrastructure with increased IM surface and detached lamellar cristae membranes as expected, independently of Mic19 re-expression (Fig. 4d,g and Supplementary Fig. 5e,j,k). Moreover, deletion of MIC19 in the Mic60$_{ProtA}$-expressing strain had a similar effect on mitochondrial architecture (Supplementary Fig. 5d,k). Re-expression of Mic19 from a plasmid in mic19Δ cells fully rescued this phenotype confirming that plasmid-encoded Mic19 is functional (Supplementary Fig. 5f,g,k). Deletion of LBS1 in Mic60$_{ProtA}$ cells (Mic60_ΔLBS1$_{ProtA}$) induced a similar phenotype as the complete absence of Mic60, the IM showing an increased membrane surface and virtually no CJs (Fig. 4e,g). Analysis of cells expressing the Mic60$_{ProtA}$ variant RW433-434DD with an inactivated LBS revealed a similar mitochondrial phenotype (Fig. 4f,g). This phenotype was also observed on Mic19 re-expression in the mic60 mutant yeast

strains (Supplementary Fig. 5d–k). Taken together, our findings indicate a requirement of Mic60 membrane binding via the LBS1 domain for maintenance of the native mitochondrial ultrastructure. Accordingly, loss of Mic60 lipid binding in mitochondria negatively affected respiratory metabolism. Even though the steady-state levels of respiratory chain (super-)complexes were similar in wild-type, mic60Δ and Mic60 LBS1-defective mitochondria as judged by BN-PAGE analysis (Supplementary Fig. 4b), we observed a considerable reduction in the enzymatic activities of complex III (cytochrome $bc_1$) and complex IV (cytochrome $c$ oxidase) (Fig. 4h,i). The reduction of the respiratory activity was thus comparable to that of mitochondria lacking the MICOS core component Mic10 (ref. 33). We conclude that lipid binding of Mic60 is crucial for MICOS integrity, mitochondrial membrane architecture and mitochondrial fitness.

## Discussion

We report an unexpected function of the contact site-forming MICOS core subunit Mic60. By directly binding and remodelling membranes, Mic60 plays a crucial role in CJ formation and shaping IM cristae. An amphipathic helix that is located between the coiled-coil and mitofilin domains of Mic60 is critical for this membrane-shaping activity. Furthermore, we show that Mic60's membrane remodelling activity is regulated by Mic19.

Earlier cryo EM tomography analyses indicated that the $F_1F_0$-ATP synthase in mitochondria of yeast and mammals forms V-shaped dimers, which assemble along the highly curved ridges of lamellar cristae, thereby stabilizing cristae curvature[14,50]. Also the mitochondrial dynamin-like OPA1 GTPase has been shown to play a crucial role in cristae remodelling and mitochondrial fusion[17,18,20,51,52]. Furthermore, a direct involvement of the MICOS component Mic10 in creating membrane curvature was recently demonstrated[33,38] indicating that Mic10 directly participates in mitochondrial membrane remodelling by oligomerization. It has been proposed that the two TM domains of Mic10 adopt a helical hairpin conformation in the IM with an asymmetric wedge shape. Moreover, several lines of evidence exist that specific phospholipids are also involved in cristae membrane organization and the generation of membrane curvature at CJs in cooperation with MICOS[34,35,45,53–57]. Our analysis now indicates that Mic60 directly contributes to membrane remodelling and mitochondrial IM architecture at CJs, but uses a mechanism different from Mic10.

Amphipathic helices are well-characterized mediators for creating and/or sensing positive membrane curvature, known for instance from amphipathic lipid packing sensor domain-containing proteins or small GTPases of the Arf and Sar1 family[47]. While they display low conservation at the sequence level, their amphipathic character is well conserved throughout evolution (Fig. 2c,d). Such helices enforce membrane curvature by asymmetrically inserting into the lipid bilayer and competing for space with the lipid head groups[58]. Often, membrane curvature by amphipathic helices is supported by rigid membrane scaffolds that impose their bent shape on the underlying membrane. For example, in N-Bin/amphiphysin/Rvs (BAR) proteins such as endophilin[59,60], an N-terminal amphipathic helix cooperates with a dimeric or oligomeric curved BAR-domain scaffold to create membrane curvature. Similar to this, the dimeric coiled-coil domain of Mic60 may assist the amphipathic LBS1 in shaping the mitochondrial membrane. Our observation that Mic60 can generate positive membrane curvature is in agreement with a role of MICOS at the rim of CJs, where positive membrane curvature needs to be stabilized (Supplementary Fig. 6). The coordination of the membrane remodelling activities of Mic60

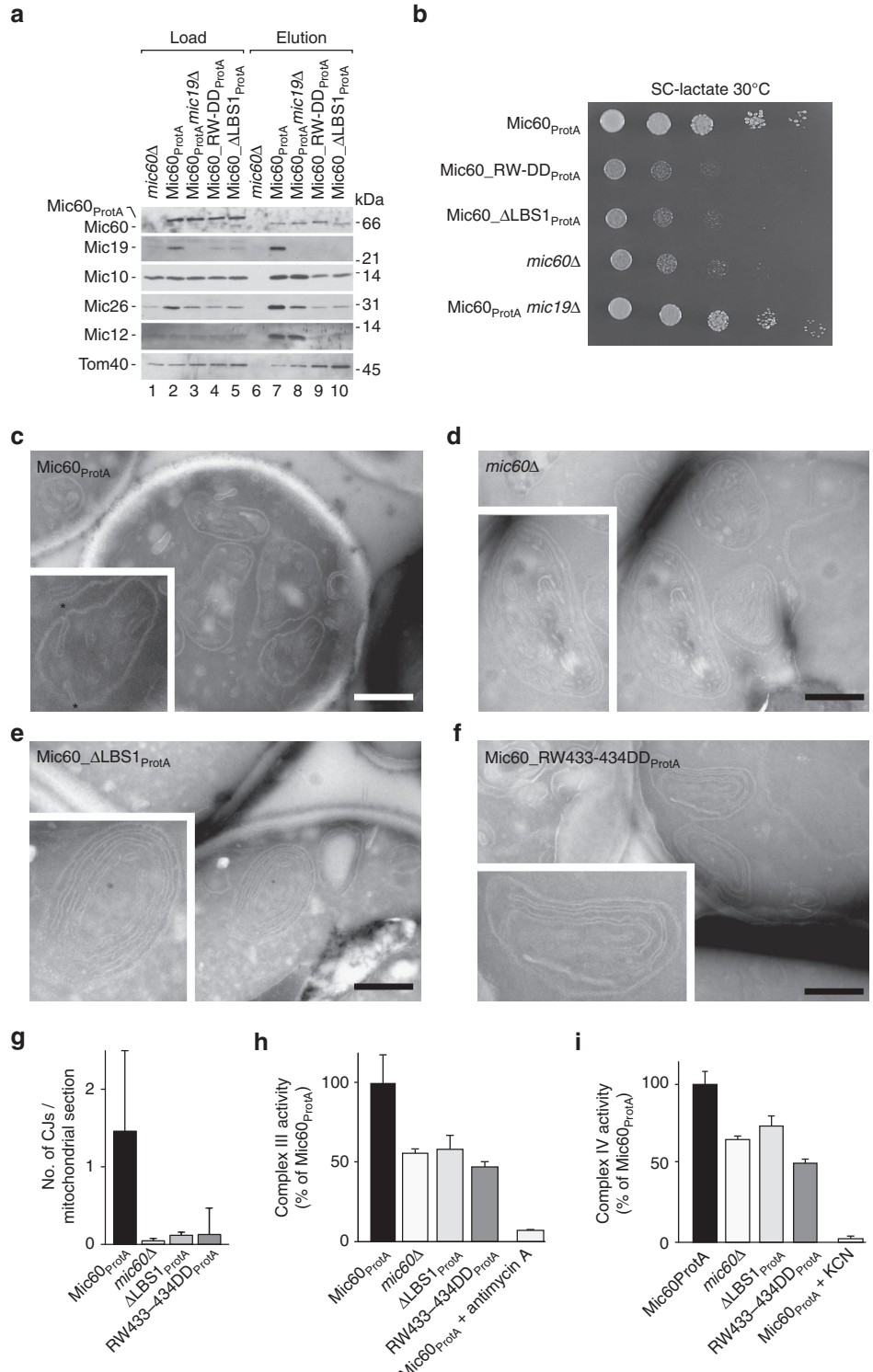

**Figure 4 | LBS1 plays a critical role in yeast mitochondria ultrastructure and function.** (**a**) MICOS integrity in the indicated yeast strains was assessed by affinity chromatography experiments from isolated, digitonin-solubilized mitochondria. Load 5%, eluate 100%; see Supplementary Fig. 7 for uncropped images, Supplementary Table 2 for used antibodies and dilutions. (**b**) Yeast growth was assessed under respiratory conditions by spotting the indicated strains on agar plates containing lactate as sole carbon (SC) source. (**c**–**f**) Electron micrographs of yeast mitochondria in ultrathin cryo sections. Characteristic cristae connected to the boundary IM via CJs are seen in Mic60_ProtA (top, left). Increased IM surface with stacked lamellar cristae sheets were detected in mic60Δ, Mic60_ΔLBS1_ProtA and Mic60_RW433-434DD_ProtA mitochondria. Asterisks mark observed cristae. Scale bar, 500 nm. (**g**) Number of CJs in electron micrographs of mitochondrial sections from Mic60_ProtA (Mic60_ProtA, n = 61) and MIC60 deletion yeast cells (mic60Δ, n = 68) as well as cells expressing Mic60 variants with deleted LBS1 (ΔLBS1, n = 39) or with LBS1 inactivated by individual amino acids substitutions (RW433-434DD, n = 42). Error bars indicate the s.d. of each data set. (**h,i**) Complex III and complex IV activity of the indicated yeast strains were measured spectrophotometrically (n = 3), error bars represent s.e.m. Mic60_ProtA mitochondria pretreated with Antimycin A or KCN served as negative controls for complex III or complex IV activity, respectively.

and Mic10 likely involves Mic12 since this subunit is crucial for the coupling of both MICOS subcomplexes[37].

In several BAR-domain proteins, such as PACSINs, membrane binding and remodelling is regulated by intramolecular auto-inhibitory action of the SH3 domain, which is released on binding to interaction partners such as dynamin[61]. Here we envisage a similar scenario for the C-terminal mitofilin domain in Mic60, which controls membrane tubulation by intramolecular interactions or binding to the CHCH domain of Mic19. It has been demonstrated before that the mitofilin domain is of crucial importance for the integrity of the MICOS machinery[40,41]. Deletion of this domain strongly impaired the co-isolation of all other subunits with Mic60 including Mic19. A possible interpretation is that a precise regulation of Mic60 membrane interaction is critical for MICOS stability. However, the mitofilin domain of Mic60 may well have distinct roles for MICOS assembly, functionality and coupling to partner protein complexes[40,41] in agreement with the strong conservation of this domain. In addition, the presence of some residual CJs in Mic60 deletion strains[21–23,32,34] argues for a second, Mic60-independent pathway for the formation of CJs.

Our findings suggest that Mic19 augments the membrane-shaping activity of Mic60. This may explain the defective cristae morphology in *mic19Δ* cells[21–23] and the observation that Mic19 knockdown in mammalian cells results in a reduced CJ diameter[39]. The CHCH domain of Mic19 mediates the contact to the mitofilin domain of Mic60, but also the coiled-coil domain of Mic19 is required for enhancing the membrane remodelling activity of Mic60. The CHCH domain of Mic19 contains conserved cysteine residues (Fig. 3a)[62] but only oxidized Mic19 carrying an intramolecular disulfide bond was found in the MICOS complex[46]. Our ITC data reveal a strong influence of the Mic19 cysteine residues on the interaction between Mic19 and Mic60. Removal of the two cysteines in Mic19 reduced the affinity for Mic60, suggesting that the intramolecular disulfide bond in the CHCH domain stabilizes the Mic60-bound conformation of Mic19, thereby modulating Mic60 membrane binding.

Taken together, our work has elucidated a so far unknown conserved function of the Mic19–Mic60 subcomplex of MICOS in generating membrane curvature at CJs, which is crucial for proper mitochondrial architecture and function.

## Methods

**Plasmids.** Codon-optimized cDNAs of Mic60 and Mic19 of *C. thermophilum* (UniProtID: G0SHY5 and G0S140, respectively) were commercially synthesized (Eurofins MWG). The genes were cloned into a modified pET28a vector encoding an amino-terminal His$_6$-tag. Mutants were produced using site-directed mutagenesis[63] and overlap polymerase chain reaction. Sequences were aligned with Clustal Omega[64] and manually adjusted.

**Expression and purification.** All constructs were expressed in *Escherichia coli* BL21 DE3 cells. Bacteria were cultured in TB medium at 37 °C to an OD$_{600}$ of 0.6 followed by a temperature shift to 18 °C. Protein was expressed for 18 h by adding 200 µM isopropyl β-D-1-thiogalactopyranoside. Cells were collected by centrifugation, resuspended in lysis buffer containing 50 mM HEPES pH 7.5, 500 mM NaCl, 1 µM DNase (Roche) and 100 µM Pefabloc (Roth), and lysed by a microfluidizer (Microfluidics). Lysates were cleared by centrifugation at 40,000g for 45 min at 4 °C. The supernatant was passed through 0.45 µm filters and applied to a 5 ml Ni-NTA column pre-equilibrated with 50 mM HEPES/NaOH pH 7.5, 500 mM NaCl, 20 mM imidazole. The column was extensively washed with this buffer. His$_6$-tagged protein was eluted with 50 mM HEPES/NaOH pH 7.5, 500 mM NaCl, 300 mM imidazole. The His$_6$-Tag was cleaved by adding His-tagged PreScission protease and dialyzed against 50 mM HEPES/NaOH pH 7.5, 500 mM NaCl overnight at 4 °C. The protein was re-applied onto the Ni-NTA column equilibrated to remove His-tagged protein. Unbound protein was concentrated and further purified by size-exclusion chromatography using a Superdex-300 column for Mic60 and a Superdex-200 column (both from GE Healthcare) for Mic19 with 20 mM HEPES/NaOH pH 7.5, 150 mM NaCl as the running buffer. Protein fractions were pooled, concentrated and flash frozen in liquid nitrogen.

**CD measurements.** Mic60 and Mic19 constructs were dialyzed against 20 mM sodium phosphate buffer pH 7.5 and 150 mM NaF and diluted to 0.2 mg ml$^{-1}$. CD measurements were performed in three replicates at 20 °C in a Chirascan CD spectrometer (Applied Photophysics).

**AUC experiments.** All measurements were performed in 20 mM HEPES/NaOH pH 7.3, 150 mM NaCl at 7 °C using an Optima XL-A centrifuge (Beckman, Palo Alto, CA, USA) and an An50Ti rotor equipped with double sector cells. Depending on protein concentration, the distribution of the protein in the cell was monitored at 230 or 280 nm. Data were analysed using the software SedFit[65]. Sedimentation velocity was run at 40,000 r.p.m. for 3.5 h, sedimentation equilibrium was performed at 9,000 r.p.m.

**Liposome co-sedimentation assay.** Folch liposomes (total bovine brain lipids fraction I, Sigma) were dried under Argon stream and solubilized in 20 mM HEPES/NaOH pH 7.5, 150 mM NaCl. Liposomes at a final concentration of 1 mg ml$^{-1}$ were incubated with 15 µM protein for 10 min at room temperature in a 40 µl reaction volume, followed by a 200,000g spin for 10 min at 20 °C (see also http://www.endocytosis.org).

**Liposome leakage assays.** We modified an existing protocol[44] as follows: 2.5 mg Folch lipids were dried under vacuum for at least 3 h and resuspended in 20 mM HEPES/NaOH, pH 7.5, 100 mM NaCl, 180 mM sucrose supplemented with 12.5 mM of the fluorescent dye ANTS and 25 mM of its contact quencher DPX (Sigma-Aldrich). The lipid suspension was incubated for 1 h at room temperature while repeatedly vortexed and then sonicated in a water bath for 30 s. Liposomes were extruded at least 11 times with 0.4 µm filters and centrifuged at 12,000g, for 10 min at room temperature. The liposome pellet was resuspended in 20 mM HEPES/NaOH pH 7.5, 100 mM NaCl at a concentration of 1 mg ml$^{-1}$. Dye release was measured in a 96-well plate with 1 mg ml$^{-1}$ liposomes containing ANTS and DPX and 50 nM protein concentration in a 100 µl reaction per well, using a FLUOstar Optima plate reader (BMG Labtech) (excitation wavelength 355 nm, emission wavelength 520 nm). As a positive control, 1% Triton X-100 (final concentration) was added. Data were normalized relative to the first data point. Averages were calculated from three independent experiments, each containing two–three parallel reactions

**ITC.** ITC experiments were performed at 10 °C in a VP-ITC (Microcal) in 20 mM HEPES/NaOH pH 7.5, 150 mM NaCl. The Mic60 concentration in the reaction chamber was around 40 µM and the Mic19 concentration in the syringe was ~450 µM. Microcal ORIGIN software was used to integrate the binding isotherms and calculate the binding parameters.

**EM.** Liposome deformation assays were done by mixing 10 µM of the Mic60 and/or 20 µM of the Mic19 constructs in 20 mM HEPES/NaOH, pH 7.5, 150 mM NaCl with Folch liposomes in a 20 µl reaction volume at final liposome concentrations of 0.8 mg ml$^{-1}$. On incubation for 30 min at room temperature, the sample was spotted onto carbon-coated grids and stained with 2% (w/v) uranyl acetate. Images were taken on a Zeiss EM910.

Liposome tubulation was quantified by counting the number of tubulated and non-tubulated liposomes (diameter ≥120 nm) from three different images (47 µm × 30 µm) Mic60 variant. If membrane tubules were observed, their outer diameter was measured at three–five positions, and at least 20 tubules per construct were quantified. The branching efficiency of each mutant was assessed by counting the number of branches per tubule for at least 20 tubules in each image field (47 µm × 30 µm).

To prepare samples for cryo-ultramicrotomy, yeast cells were fixed for 3 h with 4% (w/v) paraformaldehyde, 0.5% (v/v) glutaraldehyde in 0.1 M citrate buffer (pH and temperature adjusted to growth conditions)[12]. Samples were treated with 1% (w/v) sodium metaperiodate for 1 h at room temperature. Yeast cells were embedded in 10% (w/v) gelatin, infiltrated with 2.3 M sucrose and frozen in liquid nitrogen. Ultrathin sections were cut at −90 °C (Reichert Ultracut S, Leica) and collected on 100 mesh copper grids (Plano) coated with formvar and carbon. Sections were stained with 3% (w/v) tungstosilicic acid hydrate in 2.5% (w/v) polyvinyl alcohol. Dried grids were imaged with a transmission electron microscope (EM910, Zeiss) and acquisition was done with the iTEM software on a CDD camera (Quemesa, Emsis)

**Yeast strains and plasmids.** *S. cerevisiae* strains used in this study are derivates of YPH499 (ref. 66). Mic60$_{ProtA}$, *mic60Δ* and Mic60$_{ProtA}$*mic19Δ* strains as well as the pRS426 plasmid encoding *MIC19* have been described before[21,33]. Strains expressing the lipid-binding-deficient Mic60 variants Mic60_LBS1Δ$_{ProtA}$ and Mic60_RW433-434DD$_{ProtA}$ were generated by replacing the *MIC60* open reading frame by a cassette encoding the Mic60 variants followed by a tobacco etch virus protease cleavage site, a ProtA moiety and a *HIS3* marker gene.

**Growth of yeast cells and mitochondrial isolation.** For isolation of mitochondria, cells were grown either in YPG medium (1% (w/v) yeast extract, 2% (w/v) peptone, 3% (v/v) glycerol) or synthetic minimal medium (0.67% (w/v) yeast nitrogen base, 0.07% (w/v) complete supplement mixture (CSM)-URA amino-acid mix, 3% (v/v) glycerol, 0.1% (w/v) glucose) at 30 °C. Mitochondria were isolated by differential centrifugation. To assess respiratory growth, yeast cells were grown either in YPG medium or on synthetic minimal medium, respectively, washed in dH$_2$O and subsequently serial dilutions were spotted on to either SC lactate plates (0.67% (w/v) yeast nitrogen base without amino acids, 0.2% (w/v) synthetic complete mix, 2.5% (w/v) bacto-agar, 0.05% (w/v) CaCl$_2$, 0.06% (w/v) MgCl$_2$, 0.1% (w/v) KH$_2$PO$_4$, 0.1% (w/v) NH$_4$Cl, 0.05% (w/v) NaCl, 0.8% (w/v) NaOH, 2% (v/v) lactic acid) or synthetic minimal lactate plates (0.67% (w/v) yeast nitrogen base without amino acids, 0.07% (w/v) CSM-URA amino-acid mix, 2.5% (w/v) bacto-agar, 0.05% (w/v) CaCl$_2$, 0.06% (w/v) MgCl$_2$, 0.1% (w/v) KH$_2$PO$_4$, 0.1% (w/v) NH$_4$Cl, 0.05% (w/v) NaCl, 0.8% (w/v) NaOH, 2% (v/v) lactic acid). For EM, cells were grown for 24 h at 30 °C in synthetic complex medium[67] with 0.07% (w/v) CSM amino-acid mix including all amino acids or lacking uracil, 2% (v/v) L-lactate pH 5.0 and 0.1% (w/v) glucose. Subsequently, cells were diluted into media lacking glucose and grown overnight at 30 °C.

**Affinity purifications.** For affinity purification of ProtA-tagged proteins, mitochondria were solubilized in solubilization buffer (20 mM Tris-HCl pH 7.4, 50 mM NaCl, 0.1 mM EDTA, 10% (v/v) glycerol, 1% (w/v) digitonin, 2 mM PMSF, 1× EDTA-free protease inhibitor cocktail (Roche)). Subsequently, non-solubilized material was removed by centrifugation before mitochondrial extracts were incubated with pre-equilibrated human IgG-coupled sepharose beads (1.5 h, 4 °C). After extensive washing of the beads with washing buffer (20 mM Tris-HCl pH 7.4, 60 mM NaCl, 0.5 mM EDTA, 10% (v/v) glycerol, 0.3% (w/v) digitonin, 2 mM PMSF), bound proteins were eluted either by tobacco etch virus protease cleavage or SDS. Samples were analysed by SDS–PAGE.

**Respiratory chain complex activity measurements.** Complex III and IV activities were measured spectrophotometrically. Complex III activity measurements were carried out in complex III assay buffer (120 mM KCl, 11 µM oxidized cytochrome c (from equine heart), 5 mM malate, 5 mM pyruvate, 5 mM succinate, 0.5 mM NADH, 10 mM Tris-HCl, pH 7.4) and started by the addition of intact mitochondria that were pretreated with 10 mM KCN to inhibit complex IV activity. Reduction of cytochrome c by complex III was measured as an increase in absorbance at 550 nm. Mitochondria pretreated with 8 µM Antimycin A served as negative control. Complex IV activity measurements were performed in complex IV assay buffer (120 mM KCl, 11 µM-reduced cytochrome c (from equine heart), 10 mM Tris-HCl, pH 7.4) and initiated by the addition of Triton X-100 (0.5% (v/v) in reaction buffer) solubilized mitochondrial membranes. Cytochrome c was fully reduced with 0.5 mM dithiothreitol before addition to the assay buffer. Oxidation of cytochrome c by complex IV was measured as decrease in absorbance at 550 nm. Triton X-100 solubilized mitochondrial membranes pretreated with 10 mM KCN served as negative control.

**Data availability.** All data are available within the article and its Supplementary Information files, and from the authors on reasonable request.

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

## Acknowledgements

We thank Inge Perschil and Carola Bernert for expert technical assistance. This work has received funding from the European Research Council (ERC consolidator Grant ERC-2013-CoG-616024 to O.D.), the Deutsche Forschungsgemeinschaft (PF 202/8-1), the Sonderforschungsbereich 746 and the Excellence Initiative of the German federal and state governments (EXC 294 BIOSS; GSC-4 Spemann Graduate School to N.P. and M.v.d.L.), M.H. is a recipient of a DOC Fellowship of the Austrian Academy of Sciences at the MDC Berlin. H.R. was supported by a postdoctoral fellowship of the Peter and Traudl Engelhorn Stiftung.

## Author contributions

M.H. purified all proteins, performed the biochemical experiments and the EM experiments for analysing liposome deformation. R.M.Z., H.R. created yeast strains and performed yeast biochemical and physiological experiments. A.H.X. performed the leakage assays. S.K. and B.P. prepared and analysed cell sections. H.L. performed and analysed the AUC experiments. All authors designed research, analysed data and commented on the manuscript. M.H., N.P., M.v.d.L. and O.D. wrote the manuscript.

## Additional information

**Competing interests:** The authors declare no competing financial interests.

