## [Peer Review File · Nature Communications]

Reviewers' comments:

Reviewer #1 (Remarks to the Author):

The role of yeast MICOS complex in the cristae biogenesis has been largely investigated, but it is still unclear the molecular mechanism regulating this event. This manuscript wants to highlight the in vitro effect of the recombinant protein Mic60 variants of *Chaetomium thermophilum* in membrane deformation by using Folch liposomes, a deformation that authors suggest underlie the formation of the cristae. By using in vitro studies authors describe the tubular deformation of liposomes and identify the region LSB1 at C-terminus region as the responsible of such effect. Then, authors study how the recombinant Mic19 affects the in vitro Mic60 phenotype in the same model concluding that both proteins interact through the CHCH domain of Mic19 and the "mitofilin" domain of Mic60. Moreover they propose that Mic19 regulates the number of branch of each tubule by positively regulating the Mic60 activity.

The paper is highly valuable from the technical point of view since the purification of proteins looks optimal; however the experiments remain in vitro results that are in part the confirmation of previous in vivo (cellular models) experiments (e.g. the Mic60-Mic19 interaction). On the other hand, the new data (e.g. the Mic60 regulation by Mic19) lack in cellulo validation raising doubts on their relevance. In vivo validation of the mechanism and confirmation of the results with Mic60 variants from other yeast strains a requirement to generalize this model

Main issues.

1. In Fig. 1f the authors show representative electron micrographs of deformed liposomes after incubation with purified Mic60. It would be more informative to present a quantification of the percentage of vesicles with tubules in control (liposomes without the protein) and experiment (liposomes with Mic60), but also in the liposomes incubated with purified proteins of all the mutated constructs (Fig 2).
2. Authors suggest that the C-terminal region (here called mitofilin domain) partially inhibits Mic60 function, and that this is somehow regulated by Mic19, at least in vitro. The co-incubation of Mic60+Mic19 with liposomes displays a phenotype of liposomes tubulation with appearance of branches, concluding that Mic19 enhances Mic60 function by blocking the auto-inhibitory effect of the mitofilin domain on the Mic60 protein function. What does the branching event mean? more membrane alterations? Does it potentiate the Mic60 function or is it responsible for cristae branching? if this was the case, Mic19 overexpression shall lead to mitochondria with highly branched cristae or more cristae. However, this has never been shown and this manuscript does not show data supporting this hypothesis. Darshi et al. on the contrary show that Mic19 downregulation decrease the size of CJ in mammalian cells. Rabl et al. 2009 show that Mic60 induces branching of cristae in yeast, with no reference to Mic19. How do the authors explain these results based on their in vitro model?
3. In Fig. 2a the authors observe branched tubules in the deformed liposomes when mitofilin domain is absent from Mic60. A quantification of the percentage of vesicles with branched tubules in liposomes incubated with wt Mic60 and all the different mutants, as well as the negative control should be included. Are there fused tubules deriving from different vesicles or just branches of a single tubule?
4. Authors recognise the LBS1 region as the domain that alters membrane shape in vitro, and they identify two aminoacids as key for Mic60 function. Can the authors comment on the very low conservation of this region and the selected aminoacids among different yeast species?
5. It would be important to verify the proper folding of the different constructs.
6. In Figure 4 an in vivo experiment (like Fig 4) with the mutant Mic60sol Δ mito should be included.
7. Authors show the effect of the R572D mutant, but there is no information about the effect of the single mutant F573D on liposome binding and tubules formation. This data should be included
8. The electron microscopy experiments of yeast Mic60protAmic19 Δ mitochondria are missing.
9. The Mic19 independency of the phenotype of mutants lacking the lipid binding domain must be tested following reintroduction of Mic19 and electron microscopy.

Minor points

10. Mic60 has for sure an important role in cristae biogenesis since its deletion significantly reduce the number of cristae and cristae junctions. However, since some CJ are still present when Mic60 is deleted other proteins and/or mechanism can also regulate this process. Authors shall therefore include a note on this.

Reviewer #2 (Remarks to the Author):

The manuscript from Hessenberger et al. titled 'Regulated membrane remodelling by Mic60 controls formation of mitochondrial crista junctions' describes the membrane deformation activity of a purified aa208-691 fragment of Mic60 dubbed Mic60sol.

Mic60sol is shown to form stable dimers and to tubulate liposomes; C-terminal truncation analysis then allows the authors to identify the lipid binding site to be an amphipathic helix that is relatively well-conserved in Mic60 orthologs. The authors then go on to demonstrate that the CHCH domain of Mic19, an interaction partner of Mic60, has the ability to interact with the so-called 'mitofilin' domain of Mic60. Further data are presented to suggest that this interaction of Mic19 with Mic60 potentially regulates the membrane deforming activity of Mic60. The experiments carried out are clean, interpretable and properly controlled-for. This reviewer is not convinced, however, about the significance the authors attach to distinction between 'mostly unbranched' tubules generated by Mic60sol (line 129) and the 'increased appearance of branched tubules' using Mic60sol Δ Mito (line 138) and 'tubulation with many branches' caused by the Mic60sol-Mic19 combination (line 204). However, the possibility that the Mic19-Mic60 interaction regulates the membrane deforming activity of Mic60 is a feasible one. The authors then go on to show that the membrane-active function of Mic60 is important in MICOS complex formation. These data are somewhat weak and should possibly be included in the Supplementary Figures.

Overall, this manuscript rigorously confirms the importance of amphipathic helices in mediating membrane association by proteins and adds yet another to the long list of membrane-active amphipathic helices.

Reviewer #3 (Remarks to the Author):

Mic60 and Mic10 are the crucial components of the MICOS complex regulating cristae remodeling and crista junction formation. Yet, the direct role of Mic60 in regulating membrane remodeling is hitherto unknown. The manuscript by Hessenberger et al., entitled 'Regulated membrane remodeling by Mic60 controls formation of mitochondrial crista junctions' describes a novel and direct role of Mic60 in regulating membrane binding and remodeling. This is strongly supported by the use of in-vitro-liposome binding/tubulation assays and electron microscopy using recombinantly purified soluble domains of Mic60 derived from a thermophilic fungus (*Chaetomium thermophilum*) and liposomes. The authors further show that the interaction of the CHCH domain of Mic19 with Mic60 modulates liposome tubulation and branching. They nicely corroborate these findings by expressing analogous mutants in baker's yeast.

The authors have performed elegant liposome binding assays where they show that mixing the liposomes (obtained from bovine brain lipids) with a soluble form of Mic60 leads to formation of membrane tubules which is not observed when only liposomes or only soluble Mic60 was added. In addition, the Mic60 soluble domain is shown to exist mostly as a dimer in its native state along

with some higher oligomers as shown by sedimentation velocity analysis. Importantly, the authors have also mapped the lipid binding site in Mic60 and further mapped the amino acid residues responsible for binding of Mic60 to lipid membranes. They convincingly show that two residues within the LBS domain are critical for lipid binding and tubulation of liposomes.

Further, using ITC (isothermal titration calorimetry) studies, Mic60 and Mic19 were found to interact with each other by their mitofilin and CHCH domain respectively. It is noteworthy that Mic19 did not bind liposomes *in vitro* on its own but modulated formation of liposome tubules and branches when Mic60sol was present. For the interaction of Mic19 and Mic60, an intramolecular disulfide bond of Mic19 was necessary. The interaction of Mic60 with other MICOS proteins was reduced when lipid binding site was absent (which was mapped to amino acid residues 433 and 434). Complementation assays using baker's yeast clearly reflected this. Finally, it was shown that lipid binding mutants of Mic60 had no crista junctions leading to the formation of onion ring-like structures which basically recapitulates the phenotype of the Mic60 deletion.

Overall, this study addresses a highly interesting and relevant question and provides important insights into the way how mitochondrial crista junctions are formed. There are only minor issues to strengthen the manuscript further.

1. The general effects on membrane tubulation and branching are convincing. However, in my opinion it is needed to provide also quantitative data (membrane tubule length/branches per unit length). In particular the effect of Mic19 on branching is rather subjective and needs further quantitative support. Maybe the authors can quantify the number of branches per length of tubulated liposomes and the number of pearls on a string per length? Also the differences in membrane tubule formation of Mic60 lacking LBS1 and WT Mic60 should be quantified accordingly.
2. Regarding the electron microscopy data also the diameter of the tubes does not appear to be constant depending on the used domains/combinations. It would be good to give an overview on the obtained diameters of the tubules and to show scale bars for the close-ups of the electron micrographs as well. I feel that higher (and identical!) magnifications of the close-ups for Fig 2 and 3 as done in Fig.1f are needed.
3. Fig. 3gh: It is argued that Mic60 and Mic19 play a coordinated role to control membrane tubulation and branching (see also point 1). They argue that this is mediated by the CHCH domain harboring intramolecular disulfide bonds. However, the authors only show its binding to Mic60sol (Fig. 3e) but they do not show the effect of this domain together with Mic60sol on liposome tubulation by electron microscopy. This should be added and discussed.
4. In Suppl. Fig 3c low affinity binding of Mic19(C132S/C143S) to Mic60 is shown. Does this variant of Mic19 (or the mutated CHCH domain alone) also modulate the ability of Mic60sol to tubulate membranes and in which manner? This should be added and discussed.
5. The authors use liposomes derived from bovine brain lipids. Did the authors use liposomes with other or defined lipid composition as well? It would be interesting to see these results. Is cardiolipin is needed for the membrane tubulating activity of Mic60?
6. Mic19, 25 and 60 exist as a sub-/core-complex of MICOS. Did the author also test the effect of Mic25 on membrane tubulation in the presence of Mic60 and Mic19?
7. How could do the authors envision that Mic19 affects branching? This could be discussed more.

Reviewer #1 (remarks to the authors):

The role of yeast MICOS complex in the cristae biogenesis has been largely investigated, but it is still unclear the molecular mechanism regulating this event. This manuscript wants to highlight the in vitro effect of the recombinant protein Mic60 variants of *Chaetomium thermophilum* in membrane deformation by using Folch liposomes, a deformation that authors suggest underlie the formation of the cristae. By using in vitro studies authors describe the tubular deformation of liposomes and identify the region LSB1 at C-terminus region as the responsible of such effect. Then, authors study how the recombinant Mic19 affects the in vitro Mic60 phenotype in the same model concluding that both proteins interact through the CHCH domain of Mic19 and the "mitofilin" domain of Mic60. Moreover, they propose that Mic19 regulates the number of branch of each tubule by positively regulating the Mic60 activity.

The paper is highly valuable from the technical point of view since the purification of proteins looks optimal.

Thank you very much.

However, the experiments remain in vitro results that are in part the confirmation of previous in vivo (cellular models) experiments (e.g. the Mic60-Mic19 interaction). On the other hand, the new data (e.g. the Mic60 regulation by Mic19) lack in cellulo validation raising doubts on their relevance. In vivo validation of the mechanism and confirmation of the results with Mic60 variants from other yeast strains a requirement to generalize this model.

Main issues.

1. In Fig. 1f the authors show representative electron micrographs of deformed liposomes after incubation with purified Mic60. It would be more informative to present a quantification of the percentage of vesicles with tubules in control (liposomes without the protein) and experiment (liposomes with Mic60), but also in the liposomes incubated with purified proteins of all the mutated constructs (Fig 2).

As suggested, we quantified liposome tubulation observed in the negative stain experiments with the different recombinant proteins used in our study. These data are presented in the new Supplementary Fig. 2i and fully support our original conclusions.

2. Authors suggest that the C-terminal region (here called mitofilin domain) partially inhibits Mic60 function, and that this is somehow regulated by Mic19, at least in vitro. The co-incubation of Mic60+Mic19 with liposomes displays a phenotype of liposomes tubulation with appearance of branches, concluding that Mic19 enhances Mic60 function by blocking the auto-inhibitory effect of the mitofilin domain on the Mic60 protein function. What does the branching event mean? More membrane alterations? Does it potentiate the Mic60 function or is it responsible for cristae branching?

In response to this request, we first quantified the branches per liposome tubule obtained with Mic60 and different versions of Mic19. These data are shown in the new Supplementary Fig. 2k. To further characterize the modulation of Mic60 membrane remodeling activity by Mic19 in a more quantitative manner, we conducted a membrane leakage assay. The results

are shown in the new Fig. 3k. They reveal that addition of Mic19 greatly increases the membrane leakage (and thus membrane remodeling) activity of Mic60, whereas the isolated CHCH domain does not have such an effect. These experiments suggest that Mic19 cooperates with Mic60 during membrane remodeling. This activity apparently involves not only the Mic19 CHCH domain that mediates the docking of Mic19 onto Mic60, but also other parts of the protein. Both, our original and new data suggest that Mic19 potentiates Mic60's membrane remodeling function, potentially leading to the outgrowth of multiple tubules from reshaped liposomes.

If Mic19 potentiates membrane remodeling of Mic60, Mic19 overexpression shall lead to mitochondria with highly branched cristae or more cristae. However, this has never been shown and this manuscript does not show data supporting this hypothesis. Darshi et al. on the contrary show that Mic19 downregulation decrease the size of CJ in mammalian cells. Rabl et al. 2009 show that Mic60 induces branching of cristae in yeast, with no reference to Mic19. How do the authors explain these results based on their in vitro model?

In Bohnert et al. (2015) van der Laan, Pfanner and co-workers showed that Mic19 overexpression does not affect mitochondrial ultrastructure, likely because the excess of Mic19 protein is not integrated into the MICOS complex. This observation may explain, why an enhanced modulation of Mic60 function upon overexpression of Mic19 was not evident in vivo. Several earlier studies have shown that lack of Mic19 in mitochondria leads to a strong decrease of crista junction numbers and the formation of detached lamellar cristae in yeast. The decreased size of crista junctions observed in mammalian mitochondria upon depletion of Mic19 may be interpreted as an intermediate stage of crista junctions loss. However, it is unknown, how exactly the diameter of crista junctions is regulated. Because loss of Mic19 only moderately impairs the integrity of the remaining MICOS complex (Zerbes et al. 2016), the strong morphological changes in Mic19-deficient mitochondria implicated a specific function of Mic19 at crista junctions. Our experiments now indicate that at least part of this function may be the local stimulation of Mic60 membrane remodeling activity in the MICOS complex.

3. In Fig. 2a the authors observe branched tubules in the deformed liposomes when mitofilin domain is absent from Mic60. A quantification of the percentage of vesicles with branched tubules in liposomes incubated with wt Mic60 and all the different mutants, as well as the negative control should be included.

We added the requested quantification in the new Supplementary Fig. 2k.

Are there fused tubules deriving from different vesicles or just branches of a single tubule?

We thank the reviewer for bringing up this interesting point. Based on the maximal length of the tubules observed in EM (30 μm) and its average radius of 0.09 μm (Supplementary Fig. 2j), we can calculate the maximal membrane area of the tubule as $A = 2\pi r (r + h) \approx 8 \mu\text{m}^2$. The diameter of a typical large liposome in our assay is 1-2 μm resulting in a membrane surface $A = \pi d^2 = 3-12 \mu\text{m}^2$, which is in the same range as calculated for the tubule. Thus, full tubulation of a large liposome can result in a membrane tubule of 30 μm length. However, we cannot exclude the possibility that some of the observed long and branched tubules are derived from fusion of two smaller membrane tubules. Of note, we also observed

fragmentation of the highly branched tubular liposomes seen in the presence of the Mic60-Mic19 complex leading to the formation of very small vesicles.

4. Authors recognise the LBS1 region as the domain that alters membrane shape in vitro, and they identify two amino acids as key for Mic60 function. Can the authors comment on the very low conservation of this region and the selected amino acids among different yeast species?

The lipid-binding domain identified in this study is the most conserved region in Mic60 outside the mitofilin domain across species (see updated sequence alignment in Fig. 2c). Furthermore, membrane-binding amphipathic helices have generally a low sequence identity but are conserved in their amphipathic character (see review by Bruno Antony et al., which is now cited in the discussion), which is also the case for the LBS1 in Mic60 (see Fig. 2d). We now mention this observation in the discussion. To identify amino acid residues critical for liposome binding, we selected the conserved residues Arg572 and Phe573. From comparison with the features of known lipid-binding amphipathic helices, we reasoned that the positively charged Arg572 may represent a binding partner for negatively charged phospholipid head groups, whereas Phe573 may insert into the hydrophobic core of the lipid bilayer. Indeed, we found that both residues are important for Mic60 lipid binding.

5. It would be important to verify the proper folding of the different constructs.

We now provide CD spectroscopy data for all the different Mic60 variants used in the new Supplementary Fig. 1a. These measurements show that the overall conformation of the different protein variants is similar.

6. In Figure 4 an in vivo experiment (like Fig 4) with the mutant Mic60sol Δ mito should be included.

Soluble Mic60 lacks the N-terminal signal sequence that targets the protein to the inner mitochondrial membranes in vivo. Of note, Körner et al. (2012) generated a fusion construct composed of the soluble Mic60 portion and the N-terminal bipartite mitochondrial targeting sequence of cytochrome *b*₂. When expressed in yeast cells, this construct was targeted to mitochondria and the cytochrome *b*₂ signal sequence was processed twice as expected, releasing the large soluble domain of Mic60 into the intermembrane space. This construct was not able to complement the loss of endogenous Mic60, indicating it is not functional in vivo. We have therefore included in our analysis a yeast strain expressing Mic60 without the mitofilin domain (but containing the N-terminal mitochondrial targeting signal) from the chromosomal *MIC60* locus. In agreement with an earlier publication (Zerbes et al., 2012), this strain shows a massive loss of crista junctions in mitochondria. These new data are part of Supplementary Fig. 5 j and 5k.

7. Authors show the effect of the R572D mutant, but there is no information about the effect of the single mutant F573D on liposome binding and tubules formation. This data should be included.

We now provide the requested data in the Fig. 2h. As expected, the F573D mutant shows greatly reduced membrane binding and liposome tubulation (see also quantification in the new Supplementary Fig. 2i-k).

8. The electron microscopy experiments of yeast *Mic60protAmic19Δ* mitochondria are missing.

We included the requested data in new Supplementary Fig. 5d. The mitochondrial architecture of these this strains shows the expected phenotype with massive loss of crista junctions and formation of detached lamellar cristae structures.

9. The *Mic19* independency of the phenotype of mutants lacking the lipid binding domain must be tested following reintroduction of *Mic19* and electron microscopy.

We thank the reviewer for raising this important points. In the revised version of the manuscript we have included the analysis of *mic60* mutant yeast strains re-expressing *Mic19* from a plasmid. As shown in the new Supplementary Fig. 5, successful restoration of *Mic19* expression in the *mic60* mutant strains did NOT rescue MICOS integrity, cell growth under respiratory conditions, or mitochondrial membrane architecture. We conclude that - in full agreement with our original conclusions - the observed phenotypes are indeed caused by the comprised *Mic60* function and NOT by the reduced *Mic19* levels.

Minor points

10. *Mic60* has for sure an important role in cristae biogenesis since its deletion significantly reduces the number of cristae and cristae junctions. However, since some CJ are still present when *Mic60* is deleted other proteins and/or mechanism can also regulate this process. Authors shall therefore include a note on this.

We have included this statement in the discussion.

Reviewer #2 (Remarks to the Author):

The manuscript from Hessenberger et al. titled 'Regulated membrane remodelling by *Mic60* controls formation of mitochondrial crista junctions' describes the membrane deformation activity of a purified aa208-691 fragment of *Mic60* dubbed *Mic60sol*. *Mic60sol* is shown to form stable dimers and to tubulate liposomes; C-terminal truncation analysis then allows the authors to identify the lipid binding site to be an amphipathic helix that is relatively well-conserved in *Mic60* orthologs. The authors then go on to demonstrate that the CHCH domain of *Mic19*, an interaction partner of *Mic60*, has the ability to interact with the so-called 'mitofilin' domain of *Mic60*. Further data are presented to suggest that this interaction of *Mic19* with *Mic60* potentially regulates the membrane deforming activity of *Mic60*. The experiments carried out are clean, interpretable and properly controlled-for.

Thank you very much.

This reviewer is not convinced, however, about the significance the authors attach to distinction between 'mostly unbranched' tubules generated by Mic60_{sol} (line 129) and the 'increased appearance of branched tubules' using Mic60_{sol}ΔMito (line 138) and 'tubulation with many branches' caused by the Mic60_{sol}-Mic19 combination (line 204).

As outlined in our response to reviewer 1, we have now quantified the branching of tubulated liposomes with different Mic60 and Mic19 variants used in this study and included these data as new Supplementary Fig. 2k. We observed a similar number of branches for Mic60_{sol}ΔMito and Mic60_{sol}-Mic19 and adopted the terminology accordingly.

In addition, we have included a membrane leakage assay, which shows that the Mic19-Mic60 complex exhibits strongly increased membrane leakage activity compare to Mic60_{sol} alone. Together these data support our conclusion that Mic60-Mic19 enhances the membrane remodeling activity of Mic60 conferred by the newly identified LBS.

However, the possibility that the Mic19-Mic60 interaction regulates the membrane deforming activity of Mic60 is a feasible one. The authors then go on to show that the membrane-active function of Mic60 is important in MICOS complex formation. These data are somewhat weak and should possibly be included in the Supplementary Figures.

As explained in our response to point 9 of reviewer 1, we added *in vivo* experiments demonstrating that restoration of Mic19 expression in yeast strains expressing Mic60 variants does neither rescue MICOS complex integrity, nor respiratory growth, nor mitochondrial ultrastructure (see new Supplementary Fig. 5). These data demonstrate that the ultrastructural defects observed in mitochondria containing membrane-binding-deficient Mic60 variants are specific to the alterations of Mic60 and NOT secondary effects due to the loss of Mic19. We feel that this additional data further strengthen our conclusions and would therefore prefer to keep these data in the main manuscript.

Reviewer #3 (remarks to the authors):

Mic60 and Mic10 are the crucial components of the MICOS complex regulating cristae remodeling and crista junction formation. Yet, the direct role of Mic60 in regulating membrane remodeling is hitherto unknown. The manuscript by Hessenberger et al., entitled 'Regulated membrane remodeling by Mic60 controls formation of mitochondrial crista junctions' describes a novel and direct role of Mic60 in regulating membrane binding and remodeling. This is strongly supported by the use of in-vitro-liposome binding/tubulation assays and electron microscopy using recombinantly purified soluble domains of Mic60 derived from a thermophilic fungus (*Chaetomium thermophilum*) and liposomes. The authors further show that the interaction of the CHCH domain of Mic19 with Mic60 modulates liposome tubulation and branching. They nicely corroborate these findings by expressing analogous mutants in baker's yeast.

The authors have performed elegant liposome binding assays where they show that mixing the liposomes (obtained from bovine brain lipids) with a soluble form of Mic60 leads to formation of membrane tubules which is not observed when only liposomes or only soluble Mic60 was added. In addition, the Mic60 soluble domain is shown to exist mostly as a dimer in its native state along with some higher oligomers as shown by sedimentation velocity analysis. Importantly, the authors have also mapped the lipid binding site in Mic60 and further mapped the amino acid residues responsible for binding of Mic60 to lipid membranes. They convincingly show that two residues within the LBS domain are critical for lipid binding and tubulation of liposomes.

Further, using ITC (isothermal titration calorimetry) studies, Mic60 and Mic19 were found to interact with each other by their mitofilin and CHCH domain respectively. It is noteworthy that Mic19 did not bind liposomes in vitro on its own but modulated formation of liposome tubules and branches when Mic60sol was present. For the interaction of Mic19 and Mic60, an intramolecular disulfide bond of Mic19 was necessary. The interaction of Mic60 with other MICOS proteins was reduced when lipid binding site was absent (which was mapped to amino acid residues 433 and 434). Complementation assays using baker's yeast clearly reflected this. Finally, it was shown that lipid binding mutants of Mic60 had no crista junctions leading to the formation of onion ring-like structures which basically recapitulates the phenotype of the Mic60 deletion.

Overall, this study addresses a highly interesting and relevant question and provides important insights into the way how mitochondrial crista junctions are formed. There are only minor issues to strengthen the manuscript further.

Thank you very much.

1. The general effects on membrane tubulation and branching are convincing. However, in my opinion it is needed to provide also quantitative data (membrane tubule length/branches per unit length). In particular, the effect of Mic19 on branching is rather subjective and needs further quantitative support. Maybe the authors can quantify the number of branches per length of tubulated liposomes and the number of pearls on a string per length? Also the differences in membrane tubule formation of Mic60 lacking LBS1 and WT Mic60 should be quantified accordingly.

In response to this request, we quantified tubulation activity, tubule diameter and the branching activity for the Mic60 and Mic19 variants in this study. These data are shown in the new Supplementary Figs. 2i-k. To further study the ability of Mic19 to modify membrane remodeling by Mic60 in a quantitative manner, we included a liposome leakage assay (Michaud et al, Current Biology 2016). As shown in Fig. 3h, addition of Mic19 greatly increases dye leakage in the presence of Mic60 arguing for a cooperative action of the Mic60 and Mic19 in mitochondrial membrane remodeling.

2. Regarding the electron microscopy data also the diameter of the tubes does not appear to be constant depending on the used domains/combinations. It would be good to give an overview on the obtained diameters of the tubules and to show scale bars for the close-ups of the electron micrographs as well. I feel that higher (and identical!) magnifications of the close-ups for Fig 2 and 3 as done in Fig.1f are needed.

We quantified the tubule diameters and present the data in the new Supplementary Fig. 2j. We found that the Mic60_{sol} R572D induces larger tubule diameters than Mic60_{sol}, in agreement with a reduced membrane remodeling activity. The other Mic60 variants induced tubules of similar diameter. We also modified Fig. 2 and Fig. 3 according to the suggestion of the reviewer: All close-up views have now the same magnification allowing direct comparison of all images.

3. Fig. 3gh: It is argued that Mic60 and Mic19 play a coordinated role to control membrane tubulation and branching (see also point 1). They argue that this is mediated by the CHCH domain harboring intramolecular disulfide bonds. However, the authors only show its binding to Mic60_{sol} (Fig. 3e) but they do not show the effect of this domain together with Mic60_{sol} on liposome tubulation by electron microscopy. This should be added and discussed.

We performed the requested experiments and show the data in the new Figs. 3i, 3k, and Supplementary Fig. 2 i-k. Our results indicate that addition of the isolated CHCH domain to Mic60_{sol} led to an increased number of membrane tubules. However, in the membrane leakage assay, dye release was diminished. This argues for a cooperative role of the coiled-coil and CHCH domains of Mic19 in augmenting membrane remodeling of Mic60 (see also our response to point 2 of reviewer 1).

4. In Suppl. Fig 3c low affinity binding of Mic19(C132S/C143S) to Mic60 is shown. Does this variant of Mic19 (or the mutated CHCH domain alone) also modulate the ability of Mic60_{sol} to tubulate membranes and in which manner? This should be added and discussed.

We show the results of the requested experiments in the new Figs. 3j, 3k, and Supplementary Fig. 2 i-k. Essentially, this mutant behaved like Mic19 wt in membrane

deformation assays, suggesting that, the mutation - although reducing the binding affinity in the absence of membranes - is not sufficient to disrupt the Mic60-Mic19 interaction on membrane surfaces under our assay conditions.

5. The authors use liposomes derived from bovine brain lipids. Did the authors use liposomes with other or defined lipid composition as well? It would be interesting to see these results. Is cardiolipin needed for the membrane tubulating activity of Mic60?

In response to this request, we performed liposome binding experiments of Mic60_{sol} with Folch liposomes containing additional 15, 20 and 25% cardiolipin as shown in Fig. 1e, We observed no considerable effect for Mic60_{sol} co-sedimentation with the different liposomes. We also did not observe any clear effects of cardiolipin addition on liposome tubulation (Supplementary Fig. 2b, 2i-k). Clearly, more work is required to elucidate the exact lipid specificity of Mic60.

6. Mic19, 25 and 60 exist as a sub-/core-complex of MICOS. Did the author also test the effect of Mic25 on membrane tubulation in the presence of Mic60 and Mic19?

The Mic25 gene does not exist in *Chaetomium* and yeast (see SA Munoz-Gomez et al., Communicative and Integrative Biology, 2015). We therefore could not perform the suggested experiments.

7. How could do the authors envision that Mic19 affects branching? This could be discussed more.

In the newly included membrane leakage assay (Fig. 3k), we now show that Mic19 greatly accelerates Mic60-mediated membrane remodeling. The branching phenotype may thus be related to increased membrane remodeling activity. Our original and new data suggest that Mic19 may potentiate Mic60's membrane remodeling function. We have included this idea into the discussion as requested.

REVIEWERS' COMMENTS:

Reviewer #1 (Remarks to the Author):

I am satisfied by the revised version produced by the Authors. I only have a few minor comments

1. The membrane leakage experiments are not informative, because no positive control (i.e. detergent) is added. This shall be included to allow comparison between the effect of the Mic19-60 interaction and the maximal dequenching of ANTS.

Furthermore, a few changes in the introduction and in the reference choice shall taken into consideration:

2. Authors cite a couple of reviews from 2006 (McBride et al, Otera&Mihara) which are outdated.

Please replace with newer ones, or use the updated, 2016 reviews already cited

3. Reference 20 does not show that "cristae formation and dynamics are associated with mitochondrial fusion and fission events" but suggests that there are 2 independent pathways for cristae formation in yeast, one depending on fusion, the other on the MICOS and ATPase dimerization. The role of fusion proteins and Opa1 in particular is shown in Frezza et al Cell 2006, Glytsou et al Cell Reports 2016, Barrera et al FEBS Lett 2016.

4. A sentence on the role of CJ in apoptosis and cytochrome c release (Scorrano et al Dev Cell 2002) is warranted because of the historical role played in re-founding the field of cristae dynamics 40 years after the original Hackebrock papers on the changes in cristae shape during respiration.

5. When discussing the molecular players in the formation and maintenance of CJ, the interplay between Opa1 and MICOS in mammalian cells (Glytsou et al Cell Reports 2016, Barrera et al FEBS Lett 2016) must be introduced.

Reviewer #3 (Remarks to the Author):

The manuscript by Hessenberger et al., entitled 'Regulated membrane remodeling by Mic60 controls formation of mitochondrial crista junctions' describes a novel role of Mic60 in regulating membrane binding and remodeling. In short, all concerns were addressed well and sufficiently.

The authors have included/added the following experiments in their manuscript:

- 1) The quantification of membrane tubulation wherever required
- 2) Added EM images of membrane tubulation upon incubating Mic60sol with Mic19, CHCH domain
- 3) Added membrane leakage assay by using Folch liposomes
- 4) Done experiments showing binding of Mic60sol with liposomes containing cardiolipin

Overall, I think the manuscript is strong in its experimentation and claims and of high general interest.

Thanks a lot to reviewers 1 and 3 for their enthusiasm. Please find below the responses to the remaining concerns of reviewer 1.

1. The membrane leakage experiments are not informative, because no positive control (i.e. detergent) is added. This shall be included to allow comparison between the effect of the Mic19-60 interaction and the maximal dequenching of ANTS.

We added the requested data to Fig. 3k, the figure legend and the Methods.

Furthermore, a few changes in the introduction and in the reference choice shall taken into consideration:

2. Authors cite a couple of reviews from 2006 (McBride et al, Otera&Mihara) which are outdated. Please replace with newer ones, or use the updated, 2016 reviews already cited.

We updated the reference list and removed reviews 1, 4, 6, 11 and 64. As suggested, we now mainly refer to Parnas and Scorrano, 2016 (formerly reference 13), and Wai and Langer, 2016 (formerly reference 21).

3. Reference 20 does not show that "cristae formation and dynamics are associated with mitochondrial fusion and fission events" but suggests that there are 2 independent pathways for cristae formation in yeast, one depending on fusion, the other on the MICOS and ATPase dimerization. The role of fusion proteins and Opa1 in particular is shown in Frezza et al Cell 2006, Glytsou et al Cell Reports 2016, Barrera et al FEBS Lett 2016.

We modified the respective sentence in line 56, which is now: "It has been proposed that cristae formation and dynamics *may be connected to the mitochondrial fusion and fission machineries.*" This putative connection is a central topic of reference 20 (Harner et al. 2016, now reference 19). As justly requested, we now additionally refer to Glytsou et al. (Cell Reports 2016) and Barrera et al. (FEBS Lett 2016) - two papers that were not yet published at the time of our original submission - to further support this notion.

4. A sentence on the role of CJ in apoptosis and cytochrome c release (Scorrano et al Dev Cell 2002) is warranted because of the historical role played in re-founding the field of cristae dynamics 40 years after the original Hackebrock papers on the changes in cristae shape during respiration.

As requested, we added the following statement to the introduction (line 59): "Of note, remodeling of crista junctions to facilitate cytochrome c release from mitochondria is a key event in the induction of programmed cell death.", and cite the most relevant literature.

5. When discussing the molecular players in the formation and maintenance of CJ, the interplay between Opa1 and MICOS in mammalian cells (Glytsou et al Cell Reports 2016, Barrera et al FEBS Lett 2016) must be introduced.

Following the recommendation of the reviewer, we added this information to the introduction (line 69):

"Recent studies indicate that crista junction formation and maintenance in mammals is controlled by an intricate interplay between MICOS and the dynamin-like GTPase Opa1 (Glytsou et al., Cell Reports 2016, Barrera et al., FEBS Lett 2016)." Furthermore, we added the references Glytsou et al. (Cell Reports 2016) and Barrera et al. (FEBS Lett 2016) also to the discussion (line 336).